# Fate of Duplicated Neural Structures

**DOI:** 10.3390/e22090928

**Published:** 2020-08-25

**Authors:** Luís F. Seoane

**Affiliations:** 1Departamento de Biología de Sistemas, Centro Nacional de Biotecnología (CNB), CSIC, C/Darwin 3, 28049 Madrid, Spain; lf.seoane@cnb.csic.es; 2Instituto de Física Interdisciplinar y Sistemas Complejos (IFISC), CSIC-UIB, 07122 Palma de Mallorca, Spain

**Keywords:** duplicated neural circuits, brain symmetry, brain asymmetry, lateralization, statistical physics of neural circuits

## Abstract

Statistical physics determines the abundance of different arrangements of matter depending on cost-benefit balances. Its formalism and phenomenology percolate throughout biological processes and set limits to effective computation. Under specific conditions, self-replicating and computationally complex patterns become favored, yielding life, cognition, and Darwinian evolution. Neurons and neural circuits sit at a crossroads between statistical physics, computation, and (through their role in cognition) natural selection. Can we establish a statistical physics of neural circuits? Such theory would tell what kinds of brains to expect under set energetic, evolutionary, and computational conditions. With this big picture in mind, we focus on the fate of duplicated neural circuits. We look at examples from central nervous systems, with stress on computational thresholds that might prompt this redundancy. We also study a naive cost-benefit balance for duplicated circuits implementing complex phenotypes. From this, we derive phase diagrams and (phase-like) transitions between single and duplicated circuits, which constrain evolutionary paths to complex cognition. Back to the big picture, similar phase diagrams and transitions might constrain I/O and internal connectivity patterns of neural circuits at large. The formalism of statistical physics seems to be a natural framework for this worthy line of research.

## 1. Introduction

Statistical physics determines the abundance of different patterns of matter according to cost–benefit calculations. In the simplest cases, structures that minimize their internal energy and maximize their entropy are more likely observed than other kinds of matter arrangements. More complex scenarios require additional chemical potentials that must be optimized as well—by combining concentrations of different molecules across space, resulting in richer patterns. As external parameters (pressure, temperature, etc.) are varied, the most likely arrangements of matter might change smoothly or radically—in what we know as phase transitions. By mapping optimal structures over such relevant dimensions, we obtain phase diagrams that help us to understand what arrangements of matter to expect under distinct circumstances.

The appearance of life has been (qualitatively) described as a phase transition [1,2]. It would take place as relationships between thermodynamic potentials overcame a complexity threshold. After this, a preferred pattern of matter consists of self-replicating entities (Figure 1a) that can jump-start Darwinian evolution [3]. Subsequently, natural selection raises the bar of the thermodynamic cost-benefit requirements: stable patterns of matter in the biosphere not only need to be thermodynamically favored, but they also need to win a fitness contest (Figure 1b). As a consequence, structures within organisms often operate close to computational thermodynamic limits (e.g., of effective information processing and work extraction [2,4,5]). Despite the added layer of complexity, statistical physics remains a very apt language to describe some biological processes [6,7,8,9,10]: cell cycles can be studied as thermodynamic cycles [11,12,13], aspects of organisms might stem from an effective free energy minimization (i.e., yet another cost-benefit balance) [14], and variation along relevant dimensions (e.g., organism size vs metabolic load) determines the viability of key living structures, which can be abruptly terminated as in phase transitions [15,16]. An organism’s computational complexity, which enables its cognition, is another one such key dimension. Many essential aspects of life rely on information processing mechanisms, such as memory, error correction, information transfer, etc., resulting in a central role for computation in biology and Darwinism [2,17,18,19,20,21,22,23,24,25,26,27,28,29,30]. Additionally, once again, statistical physics plays a paramount role in computation, e.g., by setting limits to efficient implementation of algorithms [31,32,33,34,35,36].

Cognition is possible in organisms without neurons [39,40,41,42,43]; however, nerve cells are the cornerstone of cognitive complexity. Within the previous framework we wonder how to elaborate a biologically grounded statistical physics of neural circuits (Figure 1c). This should bring known thermodynamic aspects of computation into a Darwinian framework where cognition serves biological function [29,44,45,46,47,48] (eventually, to balance metabolic costs and extract free energy from an environment for an organism’s advantage). Such a theory would dictate the abundance of different kinds of circuits; or which kinds of brains to expect under fixed thermodynamic, computational, and Darwinian conditions. For example, it could tell whether “brains” are likely to have a solid substrate (such as the cortex or laptops [39]) or a liquid substrate (such as ants, termites, or the immune system [39,48,49]). We could also derive phase diagrams saying when computing units behaving as reservoirs (as in Reservoir Computing [50,51,52,53,54,55]) are more efficient [56] or, in a different context, whether redundancy or regeneration is a favored strategy to overcome neural damage [37] (Figure 1d). Matter samples that undergo phase transitions often lose or gain some symmetry [57,58] (e.g., as a glass’s grid invariance fades upon melting). Symmetry (or lack thereof) is a prominent topic for the brain [59], as it concerns redundant computations or lateralization of cognitive functions (a kind of broken symmetry). Statistical Physics has an array of tools that are ready to characterize symmetry in neural systems.

Developing a Statistical Physics of neural circuitry is a big task, but some efforts are underway [5,35,36,37,39,48,49,56,60,61,62]. To facilitate this, we can come up with prominent dimensions that capture essential aspects of partial problems. For example (besides the metabolic cost of neural circuits), the complexity of the organism’s interaction with its environment as reflected by (i) richness of input signals and (ii) generated output behavior; as well as, (iii) internal organizing principles that are needed to carry out the relevant computations. The physical scale of the investigated neural structures might be another deciding feature: a broad cortical area or a cortical column might deal more easily with excess input or output complexity than single neurons—as we will discuss. Our hope is that effective thermodynamic-like potentials might emerge associated to such relevant dimensions. Thus, we might reduce the problem to simple cost–benefit calculations again. Such coarse-graining of lesser details into effective causal tokens is a current hot topic in Statistical Physics and Information Theory [63,64,65,66,67,68,69,70,71,72]. Additionally, this is also somehow the task of the brain, as it builds efficient causal representations of its environment, or as it passes these representations around different processing centers. Perhaps, information theoretical limits to efficient symbolic encoding constrain neural wiring and communication between brain structures.

Building a statistical physics of neural circuits at large is out of our scope here, but that framework guides and inspires our research. We focus on much narrower questions: As we wonder in the title, what is the evolutionary fate of duplicated neural circuits, as they confront fixed thermodynamic and computational conditions? Given some metabolic constraints and information-processing needs, is a duplicated neural structure stable? Or is it so redundant that it becomes energetically unjustified? Can two duplicated circuits interact with each other to alter the available computational landscape? How does this affect the evolutionary path further taken by each symmetric counterpart? How is this affected by the scale of the duplicated units (e.g., are the evolutionary fates available the same for redundant neurons, ganglia, cortical columns, or layers, etc.)? Can we capture some of these aspects with simple cost-benefit tradeoffs, thus building phase diagrams to reveal evolutionary transitions?

To better ground the problem in actual biology, in Section 2 we collect examples of neural duplicities in Central Nervous Systems (CNS). The list is not exhaustive. The examples were chosen because of their prominence, because they pose open neuroscientific problems, or because we wish to explore interesting ideas about them. Sometimes the duplicity is explicit (a same circuit appears twice in our brains), sometimes it is subtler (a task is implemented twice by different structures, perhaps in different ways—thus, why this phenotypic redundancy and how is it achieved?). Throughout, we wonder what conditions (usually, of computational complexity) might support this duplicity or trigger its evolution in a phase-like transition. In Section 3, we model a simple case through naive cost-benefit tradeoffs. These result in actual phase diagrams with transitions that we can calculate explicitly. These diagrams inform us about possible evolutionary paths towards computationally complex phenotypes. These are novel results further investigated elsewhere [73]. In Section 4, we bring models, results, and examples together into the big picture exposed in this introduction.

## 2. A Showcase of Duplicated Neural Structures

### 2.1. The Two Hemispheres

The two brain hemispheres come readily to mind as duplicated neural structures. Broadly, the brain presents bilaterality with two large, mirror-symmetric halves [74,75]. Their symmetry likely stems from the body plan of bilateralians [76]—non-bilateralian ‘brains’ present other symmetries [77,78]. Part of the correspondence between the body plan and the brain is still explicit in the somatosensory and motor cortices, which contain explicit representations of body parts ordered contiguously in the brain roughly as in the body [79]. However, evidence from lesions and hemispherectomies shows that the brain’s bilateral symmetry might be redundant. Individuals who lose a hemisphere (especially young ones) can often regain control of both bilateral body sides with the halved brain alone, as well as implement other critical tasks [80,81,82,83,84,85,86,87].

A more careful examination further dismantles the appearance of bilateral symmetry in the brain [59,74,88,89]. One hemisphere motor-dominates the other, resulting in a more skilled contralateral part of the body [89]. Usually, the dominant hemisphere also houses all prominent language centers [88,90,91,92,93,94,95,96]. The areas where these centers could dwell (see next subsection) are thicker in the language-dominant side, resulting in macroscopically visible asymmetries. Primary visual cortices are fairly symmetrical, but visual processing higher up in the hierarchy is somehow lateralized [74,97,98]. The visual system is usually larger in the motor-dominated hemisphere [74].

In humans, more often the left hemisphere motor-dominates, resulting in right-handedness. The typical brain also presents developed language centers at the left, and enlarged visual cortices at the right [74]. When the dominance is reversed, the brain is mirror-symmetric to the typical one—i.e., a swap of dominance does not result in major disturbances of the neural architecture [74,88]. However, excessively symmetric brains correlate with pathologies, such as aphasia [88,99]. What are the reasons behind this lateralization of certain tasks? Some unsettled discussion exists as to whether more complex cognitive function correlates with lateralized brains [100]. Is lateralization a prerequisite for the emergence of certain traits, or does it follow from their evolution? How might an excess of symmetry (i.e., faithfully duplicated neural circuits) result detrimental [101]? These questions appear related to symmetry breaking phenomena, which are thoroughly characterized by statistical physics tools [57,58]. Can we incorporate this formalism into a computational and Darwinian framework to describe symmetry breaking in the brain?

### 2.2. The Perisylvan Network for Human Language

The discovery of Broca’s area was a milestone in the understanding of the brain [74]. It offered definitive proof that different cortical parts take care of distinct functions, and that both hemispheres are not equal. Human language is lateralized, typically with most prominent language-specific circuitry at the motor-dominant hemisphere. This circuitry is located around the Sylvian fissure [90,91,92,93,94,95,96]. It includes the Broca and Wernicke areas, among others, and abundantly interfaces with motor and auditory cortices. The language-dominated hemisphere lacks these structures and the thick wiring connecting them. Both hemispheres are more symmetric at birth [88,96,102]. Evidence suggests that similar circuits exist at either side in newborns, and that both react to speech as soon as day 2 (even with a preponderant reaction on the right side) [102]. Presumably, only the circuits at the dominant side mature into the fully lateralized perysilvan langauge network. Some adult brains are less lateralized, developing seemingly symmetric circuitry at both sides. Such brains more often present language pathologies [88,99]. This strongly suggests that a duplicity of language circuits is counterproductive. Might this be due to a conflicting interference between two candidates for language production [101]?

Thus, normal language development suggests that it is convenient to lose some of the innate duplicity. However, clinical cases of language recovery after hemispherectomy or injury of the dominant side suggest that enough redundancy can persist in the dominated hemisphere. These studies show that children who lost their matured language centers can grow them anew in the opposite side, and that this is easier the earlier that the intervention or injury takes place [80,83,84,85,86,87,103]. A mainstream explanation of this capability posits that the brain is more plastic at younger ages, thus potential to reconstruct language remains. This plasticity would be gradually lost, eventually preventing fully functional language from regrowing in the dominated hemisphere. This would suggest that the duplicity of neural circuitry that is related to language is not realized, but potential. However, when fully functional language develops after hemispherectomy or injury, the corresponding centers do not establish themselves in arbitrary places, but in the corresponding Wernicke, Broca, etc. territories of the dominated sides. Some duplicity, a blueprint of the missing circuits, must exist to guide this process. How is this latent duplicity balanced to prevent unhealthy interferences in healthy brains?

### 2.3. Internalizing the Control of Movement

Rodolfo Llinás suggested that the evolutionary history of the CNS is that of the progressive “encephalization” of motor rhythms [104,105]. We can find living fossils of some stages in a range of species, including ours. The earliest circuits for motor control, still present in Cnidarian and others, dwelt right under the skin [77]. Some arthropods have decentralized ganglia to coordinate their motion [106,107]. Much simpler ganglia still take care of fast reflexes in most other species [108]. As we progress towards more complex brains, movement coordination is centralized and layers of control are added. In reptiles, birds, and mammals, the brain stem coordinates several stereotyped behaviors—e.g., gait, chewing, breathing, or digesting [109]. The voluntary aspect of these and other, more complex tasks originates at subcortical centers or motor cortices.

In this process, new structures take over tasks previously managed by simpler neural centers. Whenever this happens, some overlap in function exists. Eventually, the older control structures become controlled by the newer ones. They might lose some of its complexity (e.g., ganglia controlling reflexes in mammals). In any case, they enable a hierarchical control exerted from the top. This requires that a dialog be successfully established across levels. The brain stem plays a paramount role in this sense: it works as a Central Pattern Generator [105,109,110] that translates signals from higher up centers into salient electric waveforms that activate motor neurons in an orderly manner—thus establish, e.g., gait rhythms. Transition between different gaits (walk, trot, gallop, etc.) is discrete, as is the transition between the patterns generated at the brain stem.

Throughout the motor control hierarchy, the resolution of simpler tasks coarse grains them so that they become building blocks for the next level. Similar phenomenology is currently under active research in information theory, through the study of coarse grained symbolic dynamics [63,64,65,66,67,68,69,70,71,72]. What triggers the emergence of new layers of control in this hierarchy? Is this externally motivated—e.g., because a new range of behaviors becomes available and more complex control is needed? Or internally—e.g., because an increased computational power of the CNS prompts a reorganization [111]? The statistical physics of coarse grained symbolic dynamics might shed some light in these questions, as well as research on robot control [109].

### 2.4. Place and Grid Cells—A Twofold Representation of Space?

Both grid and place cells represent space [112,113,114]. Grid cells are located in the medial entorhinal cortex. They build an exhaustive representation of the available room [114,115,116]. Each grid cell encodes space periodically, such that its receptive fields are the nodes of a grid with fixed spatial period (Figure 1e). Different grid cells use different periodicity and phase, such that each point is uniquely encoded by the spiking of confluent nodes. Place cells are located in the Hippocampus. Instead of responding periodically over space, they code specific, individual spots—a single place. They can also code for salient elements in a landscape or more extended areas—e.g., the length along a boundary.

Why this duplicitous space representation? Is it needed, convenient, or redundant? Did it emerge as each structure specialized in some specific aspect of spatial coding? In this sense, grid cells have been associated to path integration [114,115,116,117,118,119,120], indispensable to keep track when external cues are missing. And place cells relate to episodic memory, memory retrieval, and consolidation of trajectories [119,121,122]. They also integrate non-spatial modal information [123].

We entertain a complementary possible origin about the necessity of a twofold space representation. Our hypothesis is compatible with other jobs for both cell types. In [124], movement planning is solved as a constraint satisfaction problem. Networks of spiking neurons are extremely apt at quickly finding great solutions to such problems [125,126], but only if we manage to encode the constraints in the neural network topology. Different neurons in the network become the embodiment of causal variables of the problem. Their firing represents different combinations of the problem’s variables—i.e., candidate solutions for the constraint satisfaction. As neurons repress or activate each other (as dictated by the network’s wiring), they test candidate solutions against the problem’s constraints. To encode movement planning like this, a two-fold representation is needed [124]: first, an exhaustive representation of the available room; second, an encoding of relevant elements that act as constraints (e.g., walls that cannot be traversed, goals that offer a reward when reached). The first representation acts as a virtual space upon which external constraints or internalized goals can be uploaded. Both representations interact to generate ordered spiking patterns representing candidate trajectories. Optimal paths satisfy the problem’s constraints, avoiding obstacles and reaching goals.

It is tempting to assign grid cells the role of a virtual, all encompassing space representation. Place cells could then impose constraints derived from internal goals or external cues. In [124], the virtual space is a square grid with one neuron per grid position. This requires ∼*N* neurons to encode the *N* discrete locations. It is more efficient to represent the *N* sites with a binary code, which uses around ∼log(N) neurons. This is achieved by neurons coding position periodically, just as grid cells do. Thus, grid cells build efficient representations of the available room [114]—disregarding of their role in our constraint satisfaction scheme.

Regarding place cells: they respond to landscapes, to stimuli in the environment, to non-spatial features, such as odors, directionality, etc. [123,127,128]. Their firing can change as a response of environment manipulations [122,127,129]. This remapping can happen due to the change of a location’s relevance in an ongoing task [130]. Firing can also be modulated by specific behaviors (e.g., sniffing) at a location [131]. Place cells have also been assigned a predictive role, as they engage in mechanisms relevant for reinforcement learning [132,133]. For example, consecutive place cells representing a path are known to fire sequentially before that path is taken—thus preplaying it [119,134,135]. An already traveled path is also replayed once a destination is reached, potentially allocating rewards [119]. The hippocampus contains populations explicitly encoding goals and rewards [136,137]. All of these features are ideal for hippocampal cells (and specifically places cell) to represent constraints (objectives, destinations, places to avoid, etc.) to be uploaded onto the virtual room representation of grid cells, just as in [124]. This hypothesis is a beautiful way forward to understand our navigation system, and how its different parts come together as a distributed algorithm. Some ongoing research seems to point in this direction [115,132,133].

### 2.5. Somatosensory and Motor Cortices

Evidence from extant animals indicates that the earliest neocortical structures dealt mostly with sensory input [138,139], with prominent roles for visual, auditory, and olfactory modes. Somatosensory cortices evolved earlier than their motor counterparts. The most evolutionarily ancient, extant mammal species (e.g., opossums) seem to lack a motor cortex [140]. The corresponding (rather poorly developed) motor functions are handled by the somatosensory cortex itself [138,139,140] or by thalamic centers [138,141]. A primary motor cortex appears with placental mammals [138,139,142]. The number and complexity of devoted motor cortical areas grows as we approach our closest relatives [138,139]—notably, first, due to the motor-visual integration demanded by tasks such as reaching or grasping [138,139,143,144].

If we want to build an effective controller of a complex system (e.g., a cortex to handle a body with many parts), it is a mathematical necessity that the controller has to be a good model of the system itself [145]. We might think of somatosensory areas as the model within the larger (brain-wide) controller. This suggests that the evolution of somatosensory and motor areas must be intertwined, and that somatosensory maps should predate complex motor control. However, why does a dedicated motor cortex appear eventually if motor control could already be handled by somatosensory centers [138,139,140]? What evolutionary pressures prompted this emancipation of the motor areas? Does the motor cortex rely on the somatosensory cortex for the modeling needed for control? Or do motor areas develop their own modeling? If so, how does this differ from the representation in somatosensory areas? How much modeling redundancy exists in both these physically separated cortices?

We have chosen the somatosensory–motor axis, but we might also ask similar questions about redundancy in somatosensory areas. They accommodate coexisting parallel representations of the same body parts [138,139,146]. What prompted this abundance of duplicated circuits? How did they emerge? Is there something about the computational nature of the task (body representation and motor control) that favors such coexistence of redundant circuits?

### 2.6. Reactive versus Predictive Brains

In [147], Andy Clark defends the *predictive brain hypothesis*—also studied as predictive coding [148,149,150,151]. According to this view, complex brains do not stay idle, awaiting external inputs. Instead, they embody generative models that continuously put forward theories about how external environments look like. Representations built by those generative models would flow from high in a cognitive hierarchy towards the sensory cortices. There, they would be confronted with external signals captured by the senses. These signals would correct mismatches and further tune those aspects of the generated representation that were broadly correct. In predictive brains, input signals would still be necessary, but not as the main actors that build complex percepts. Instead, they would merely serve as an error correction mechanism to contrast the generated models. What flows upwards in the hierarchy are the discriminating errors that need to be discounted in the generated representations [149,150,151,152,153,154,155].

Advanced visual systems are a favorite example [148,149,150]. Again, higher cognitive centers elaborate hypotheses about a current scene. The generated representation flows from cortices that suggest shapes and locations for distinct objects, down into primary visual cortices that hypothesize about the location and orientation of these object’s edges. There, the input stream is discounted, making no correction if the hypotheses flowing top-down are correct. Otherwise, discriminant errors (signaling, e.g., misaligned edges, mismatching 3-D shapes, etc.) make their way bottom-up until they tighten the right screws across the visual hierarchy. A similar confrontation would happen at each interface between levels in this hierarchy. Some empirical evidence supports this view of complex visual systems [149,150,153].

Some Machine Learning approaches draw inspiration from the predictive brain hypothesis [156,157,158]. However, historically, most mainstream Artificial Neural Networks for vision or scene recognition work in a reverse fashion [159,160]. Visual representations are generated from scratch as the input (processed sequentially from the bottom up) reaches the distinct layers. We term this a reactive brain. The wiring of reactive brains is simpler than that of predictive brains. The former only need to convey information in one direction, while the later need to allow two: generative model outputs flow top-down and correcting errors climb bottom-up. If cognitively advanced brains are predictive, rather than reactive, they need to harbor duplicated circuitry to support the bidirectional flow. Interesting questions arise:Is there a threshold of cognitive complexity beyond which predictive brains always pay off? Protozoa reacting to gradients for chemotaxis hardly need a model of possible distributions of chemicals. The ability to dream in complex brains reveals the existence of generative models. When did they become favored?What environments are more likely to select for predictive brains? Picture a scene that varies wildly over time, often showing new inputs that a brain could hardly imagine based on previous experience. The cost of correcting hypotheses from generative models might be unbearable. Building representations anew from each incoming input (just as in deep neural networks) might be cheaper. Similarly, a pressure to anticipate the environment (e.g., to react to looming dangers) must be important—otherwise, reactive brains would do just as well and are cheaper.How did the duplicated circuitry needed for predictive brains emerge? Was a same system duplicated and reversed? Did, instead, a structure dedicated to error propagation grow over a previously existing scaffold, reversing the flow of information as it expanded? Is there a site in a cognitive hierarchy in which the predictive stance is more easily adopted and, hence, the needed circuitry expands from there? Or did both flow directions of predictive brains evolve simultaneously since early? This later possibility demands that even simple brains allow the predictive way of working, which might be possible [161,162,163,164]. Note that complex brains might also work in a reactive manner if needed—as reflexes do in advanced nervous systems [108].

We suspect that some answers to these questions might be constrained by information theory limits to channel capacity and error correction. Numerical evaluations of performances of reactive versus diffusive circuits under different computational conditions seems affordable. This would help us to build phase diagrams and characterize transitions between these brain classes.

### 2.7. The Cortical Column

Rather than a duplicated neural structure, we look at a multiplicated one. The Cortical Column (CC) [38] (Figure 1f) has been proposed as a basic computing unit of the mammalian brain [165,166]. Hypothetically, evolution stumbled upon this complex circuit so apt for advanced cognition that it was “manufactured” *en masse*, eventually populating the whole neocortex [165,166]. This story is appealing, but there is no consensus around it [167].

Single CCs have been used as a model reservoir since the inception of Reservoir Computing [50,51,52,53,54,55]. A reservoir is a highly non-linear dynamical system that projects input signals into large-dimensional spaces. In them, different input features become easy to separate from one another. Within this framework, a CC would not be task-specific, but rather work as a generic reservoir that separates potentially meaningful features of arbitrary inputs. Other, dedicated circuits would be tasked with selecting the relevant information from among all of the insights that a CC makes available. This would make CCs very versatile computing units, as they can be adapted to different purposes and even implement different functions simultaneously [50].

Evidence of computing principles compatible with reservoir-like behavior in CCs is scarce [56,168,169,170,171]. In contrast, there is abundant evidence of task-specificity. CCs morphology and abundance varies across the cortex, likely influenced by their computational idiosyncrasies. The mouse whisker barrel cortex is an extreme example. Individual CCs of this cortex have grown and thickened through evolution to take care of the somatosensory processing of each whisker [167,172]. The purported versatility of CCs (including this capacity to commit to a task and change morphology over evolutionary time), make them an interesting candidate, as a computing unit, to test the ideas exposed in the Introduction [56]: What is the steady shape and dynamics of a CC under fixed computational and thermodynamic (energetic, input entropy, etc.) constraints? Under which conditions does a CC remain reservoir-like—thus task-versatile? What conditions, instead, prompt their evolution towards task-specific forms? What might trigger their multiplicity across the neocortex? Theoretical advance seems plausible through relatively simple computer simulations. From the empirical side, CCs across species and cortical areas seem to be real-life versions of the proposed experiments.

## 3. Simple Models for a Complex Research Line

The just reviewed examples show promising intersections between statistical physics, information theory, and the evolutionary origin and fate of duplicated neural structures. In this section, we develop specific models that capture tradeoffs of circuit topology and computational complexity. Through these, we build phase diagrams—explicitly in Section 3.1 and qualitative in Section 3.2.

### 3.1. A Naive Cost-Efficiency Model of Duplicated Circuitry for Complex Tasks

We aim at building the simplest model that still captures some aspects which, we think, determine the stability of duplicated versus single neural structures. Therefore, we make mathematical choices that simplify our calculations, then look at what such minimal model can teach us. Alternatively, less simple models will be discussed elsewhere [73]. The essential aspects that we wish to capture are:A neural structure garners a fitness advantage by successfully implementing some computation. This results in a cognitive phenotype that makes the organism more apt at navigating its environment, mating, obtaining food, etc; thus, securing more energy to sustain its metabolism and, eventually, increasing progeny. A duplicated neural structure can result in computational robustness (i.e., more reliable cognition), even with faulty components [173,174].Computation is costly. A circuit’s physical structure (neurons, synapses, etc.) has a material and metabolic stress. Signaling between neurons has a high energetic toll [175], thus a circuit’s cost grows with its wiring complexity, or if connections become too long [139]. Duplicated structures would pay twice these costs, resulting in a pressure against redundancy [176,177].Coordinating duplicates is also costly. It often requires additional structures (with its associated costs) to integrate the many outputs. If missing, failure to coordinate can become pathological [178]. If the duplicated structures are far apart (e.g., at bilaterally-symmetric, distant positions in each hemisphere), output integration would pay the cost of lengthy connections as well. This results in further pressures against duplicated circuits [177,179,180].

Assume a complex neural structure composed of a number of subcircuits or submodules. We will study whether it pays off to duplicate such a structure in two different scenarios:In the first (uncooperative) scenario, each subcircuit implements a task different and independent from the tasks of other submodules. The implementation of each task results in a fitness benefit, independently of whether the other subcircuits function correctly.In the second (cooperative) scenario, the neural structure has been presented a chance to evolve a more complex cognitive phenotype. Its successful implementation reports a large fitness benefit—but only if all tasks are correctly implemented. Thus, while each subcircuit still performs different computations, they are no longer independent.

These two scenarios roughly model (i) evolutionary preconditions that are independent of each other and (ii) the process that integrates them into a new, emerging cognitive phenotype.

If duplicated, each copy of the neural structure contains the same number of submodules. Equivalent subcircuits at either structure solve a same task, so a coordination cost ensues. In return, as remarked above, computation might be more robust [173,174]. These potentially duplicated structures might be bilateral counterparts or other, arbitrary, duplicated circuits not obeying to bilateral symmetry. For simplicity, let us assume bilaterality and label these structures $S^{L}$ and $S^{R}$ (for left and right). We will say that a phenotype has “bilateral symmetry” if it favors structure duplicity, and that it is “lateralized” if only one is preferred. This is just a convenient notation—our analysis remains valid for arbitrary (non-bilateral) duplicates.

Let $K^{0}$ be the number of tasks available—so that $S^{L/R}$ consist of $K^{0}$ submodules. Let us assume that each structure has an active number of submodules $0\leq K^{L/R}\leq K^{0}$. Active modules will enter the cost-benefit calculation, inactive ones will not. The probability of selecting a module at random and finding it active is $\kappa^{L/R}\equiv K^{L/R}/K^{0}$. Furthermore, subcircuits are unreliable—each of them computes incorrectly with probability ε.

With this, let us write costs and benefits in the uncooperative scenario. In it, a benefit *b* is cashed in by each independent, successfully implemented subtask. Thus, a global benefit reads:(1)$$\begin{array}{ccc} B & = & b\left[\left(1-\varepsilon \right)\kappa^{L}\left(1-\kappa^{R}\right)+\left(1-\varepsilon \right)\left(1-\kappa^{L}\right)\kappa^{R}+\left(1-\varepsilon^{2}\right)\kappa^{L}\kappa^{R}\right]K^{0}. \end{array}$$

Within square brackets we have, for each subtask, the probability that only the left submodule is active and computes correctly, the probability that only the right submodule is active and computes correctly, and the probability that both submodules are active and at least one computes correctly. The activation of both submodules for a same task entails a coordination cost:(2)$$\begin{array}{ccc} C & = & c\kappa^{L}\kappa^{R}K^{0}. \end{array}$$

We further assume a fixed cost for each active module independent of coordination:(3)$$\begin{array}{c} \hat{C}=\hat{c}\left(1-\varepsilon \right)\left(K^{L}+K^{R}\right)=\hat{c}\left(1-\varepsilon \right)\left(\kappa^{L}+\kappa^{R}\right)K^{0}. \end{array}$$

We made this cost grow with the module’s efficiency (i.e., fall with ε). For simplicity, we assume a linear dependency. Other, non-linear alternatives will be discussed in [73]. The resulting utility function (normalized by $K^{0}$) reads:(4)$$\begin{array}{ccc} \rho & = & b\left(1-\varepsilon \right)\left[\kappa^{L}\left(1-\kappa^{R}\right)+\left(1-\kappa^{L}\right)\kappa^{R}+\left(1+\varepsilon \right)\kappa^{L}\kappa^{R}\right]-c\kappa^{L}\kappa^{R}-\hat{c}\left(1-\varepsilon \right)\left(\kappa^{L}+\kappa^{R}\right). \end{array}$$

We now study the cooperative scenario. In it, the different tasks need to become interdependent to earn the fitness reward. Let this reward be $bK^{0}$, so it grows with the complexity (in number of modules involved) of the phenotype. Multiplying by the likelihood that all tasks are successfully implemented:(5)$$\begin{array}{ccc} \tilde{B} & = & bK^{0}\cdot {\left(1-\varepsilon \right)}^{K^{0}}{\left[\kappa^{L}\left(1-\kappa^{R}\right)+\left(1-\kappa^{L}\right)\kappa^{R}+\left(1+\varepsilon \right)\kappa^{L}\kappa^{R}\right]}^{K^{0}}. \end{array}$$

The costs remain the same, so the second utility function (normalized by $K^{0}$) reads:(6)$$\begin{array}{ccc} \tilde{\rho } & = & b\cdot {\left(1-\varepsilon \right)}^{K^{0}}{\left[\kappa^{L}\left(1-\kappa^{R}\right)+\left(1-\kappa^{L}\right)\kappa^{R}+\left(1+\varepsilon \right)\kappa^{L}\kappa^{R}\right]}^{K^{0}}-c\kappa^{L}\kappa^{R}-\hat{c}\left(1-\varepsilon \right)\left(\kappa^{L}+\kappa^{R}\right). \end{array}$$

Always seeking simplicity, we assumed a linear weighting of costs and benefits. Hence, parameters *b*, *c*, and $\hat{c}$ act as external biases and correcting factors to homogenize units. More rigorously, we should optimize costs and benefits independently, as in Pareto Optimization [181,182,183]. However, this formalism maps back to statistical physics, and phase diagrams result from the utility functions above [183,184,185,186,187].

Take b=1 without loss of generality, and a fixed cost $\hat{c}=0.1$. Optimizing Equations (Equation 4) and (Equation 6) we obtain the phase diagrams from Figure 2. They tell us what structures to expect depending on a series of metabolic (energetic) costs linked to successful information processing. A first important result is that neither equation admits graded solutions: either both structures are kept, or one, or none. This naive account does not support a backup structure that takes care of partial computations. If each individual subtask contributes a fitness independent of all others (Equation (Equation 4)), then it always pays off to implement either one or both structures (Figure 2a). This is not the case for the emergent phenotype (Equation (Equation 6)): in its phase diagram (Figure 2b) for a broad region of parameters, it never pays off to build any structure at all.

Superposing both phase diagrams (Figure 2c) reveals evolutionary paths for transitions into the novel, complex phenotype. For a broad region, the complex phenotype cannot be accessed (labeled “No emergence” in Figure 2c). For the rest of the diagram, there is a direct evolutionary path towards the emergent phenotype. This includes cases in which bilateral structures remain in place (“Bilateral to bilateral” in Figure 2c) and cases in which a single structure (already optimal for the subtasks) implements the emergent phenotype on its own (“Lateral to lateral” in Figure 2c). Notably, this naive tradeoff never shows a bilateral structure that becomes lateralized as the complex phenotype emerges. To observe such a transition we would need to move around the phase diagram—i.e., the new emergent phenotype must change ε or *c* (blue curves in Figure 2c). This might happen, e.g., if the complexity of the emergent phenotype imposes a higher reliability (lower ε) for all the parts, or if coordination becomes more costly—e.g., because less discrepancies are tolerated).

### 3.2. The Garden of Forking Neural Structures

In an outstanding, broad region of the phase diagram (“Lateral to bilateral” in Figure 2c), it is favored that a single structure becomes duplicated when a high fitness can be gained by the complex, emergent phenotype. Remind that ‘lateral’ and ‘bilateral’ are just convenient labels—our results concern any neural structure afforded such an evolutionary chance. Based on our analysis, the duplication of extant structures appears as a reliable evolutionary path. Perhaps it was followed by some of the cases reviewed in Section 2. Empirical observations have indeed recently suggested that path duplication is a key mechanism for the unfolding of brain complexity [188], and that it offers buffering opportunities similar to those presented by duplicated genes [189,190]. In the mechanism of gene duplication, a copy remains faithful and functional, while the other explores the phenotypic landscape, often uncovering new functions. Might duplicated neural structures wander off in a similar manner? What might their evolutionary fate become as learning or Darwinian selection press on? How does this fate depend on the task at hand? For example, how does it change with the complexity of the input signals or of the desired output? Additionally, with the complexity of the input-output mapping? In this section we introduce a *Gedankenexperiment* to gain some insights about further evolution of duplicated neural structures. We will also attempt to fit examples from Section 2 into this qualitative framework. Necessarily, our conclusions are speculative.

Take a single neuron that implements a specific function. It receives weighted inputs from some sources (potentially including itself, thus recurrence is allowed). Its output feeds into a set of ‘actuators’ to produce some function, which also results from a weighted sum of the neuron’s output. Now, let us precisely duplicate this neuron (same weights at the in-, self-, and out-synapses) and let us add a few random links between the (now two) neurons (Figure 3a). Next, let us feed them a stream of inputs while implementing some plasticity rule (e.g., spike-timing dependent plasticity [191]). Of course, let this plasticity be influenced by the fitness of the resulting behavior (via the actuators). Correct implementation (e.g., of the previous, or of a new, emerging cognitive phenotype) should reinforce the connections that produced it. Wrong implementations should penalize the corresponding weights.

What happens to the duplicated neurons?
Is one of them lost, thus reverting to the original configuration? Subsequently, duplicating this structure was never favorable in the first place. Do they help each other instead, achieving a more robust computation?Do they become respectively specialized in pre-processing the input and producing elaborated outputs (Figure 3b)? This reminds us of the specialization of somatosensory and motor cortices, or of Wernicke’s and Broca’s area—rather processing input and producing output syntax, respectively.As the neuron is duplicated, the fitness landscape of computational possibilities changes. For example, it might become feasible to implement the original function in a hierarchical fashion, as we have seen in motor control. Might the neurons arrange themselves in a ‘controller-controlled’ architecture (Figure 3c)? Might them, instead, take care of different subsets of the function—becoming effectively uncoupled? Or might they unlock previously unavailable phenotypes, thus expanding the computational landscape (Figure 3d)?

Two duplicated neurons might be too simple a system and some options might be locked. What happens if, instead, the duplicated neural structure is a small ganglion, a complete cortical column, or a whole patch of cortex? What if it is a large, complex structure functioning as a phenotype module (e.g., the whole fusiform face area)? Do new evolutionary paths become available beyond some complexity threshold of the duplicated substrate? Are some configurations easier to achieve for simpler circuits, and others favored for more complex structures? Or, does the landscape of possibilities remain roughly similar to the one for the duplicated spiking neuron?

Numerical experiments to shed some light on these questions are underway. Meanwhile, we single out dimensions that are relevant for the problem. Final evolutionary paths will depend on the specific task at hand, but three quantifiable aspects might stand out: (i) the complexity, or intricacy, of input signals ($K_{IN}$); (ii) the complexity, or richness, of the sought behavior ($K_{OUT}$); and, the complexity of the input-output mapping ($K_{MAP}$). These quantities suggest a morphospace, where we can locate some of the neural structures discussed above (Figure 3e). Morphospaces remind us of phase diagrams. They are less rigorous, but still useful tools to contemplate possible morphologies or designs [192,193,194,195,196,197,198,199]. They sort out real data or architectures that are produced by models with respect to some magnitudes, thus helping us to compare structures to each other.

While speculative, we think that our example is a worthy exercise. We separated two large volumes of morphospace for reactive vs. predictive brains. The rationale for these locations is that: (i) predictive brains should be mainly guided by input complexity, as their output (a reaction) lags behind; (ii) predictive brains should be able to produce a wide array of representations, so that minor corrections sufice to tune them. Let us ellaborate on these reasons.

Reactive brains often reduce the input richness into categorical responses. Some neural circuits fall in this region: (i) the brain-stem working as a Central Pattern Generator takes as input the motor behavior planned in higher cortical areas and reduces it to sequences of stereotypical patterns that motor neurons easily understand. (ii) Most animal communication systems (far from the human faculty of language) also reduce a range of possible scenarios into categorical responses that (opposed to human language) must be communicated without ambiguity [199,200,201,202].

Rather predictive brains require little input complexity to prompt varied cognitive responses. Some extremely simple circuits implement predictive dynamics [161,162,163,164], but we assume that, in general, the I/O mapping ($K_{MAP}$) of predictive brains is more complex, as it can be context dependent. At the core of a predictive brain there are generative models that produce a range of possibilities being constantly evaluated and corrected [147]. They must present some recursivity, thus the interaction of the model’s state with itself would add to $K_{MAP}$.

For three more examples, we postulate similar and fairly high input and output complexity ($K_{IN}$∼$K_{OUT}$) and increasing mapping complexity ($K_{MAP}$). These are the representation of geometric space (with smaller $K_{MAP}$), sensory-motor maps (intermediate $K_{MAP}$), and human language (largest $K_{MAP}$). Most likely, the navigation and sensory-motor systems of different species present a range of $K_{MAP}$. In any case, the sensory-motor and language systems suggest that balanced $K_{IN}$∼$K_{OUT}$ more easily leads to duplicated structures specializing, respectively, in input pre-processing and rich output generation—perhaps above a certain $K_{MAP}$ threshold.

## 4. Discussion

Thermodynamics dictates the abundance of distinct matter phases through straightforward cost-benefit calculations [183,184]. In the simplest case, those patterns that minimize internal energy and maximize entropy are more probable. These calculations become more complex as other (e.g., chemical) potentials become relevant. A transition to life and Darwinian evolution [1,2,3] raises the cost–benefit stakes: matter patterns need to be thermodynamically favorable and win an evolutionary contest. Interestingly, the formalism of statistical physics remains effective to describe phenomenology across biology and cognition [6,7,8,9,10,11,12,13,14,31,32,33,34,35,36]. The later is enabled by expanded computational complexity and allows organisms to increasingly integrate and predict environmental information [2,17,18,19,20,21,22,23,24,25,26,27,28,29,30].

A thermodynamically viable and evolutionarily successful organism must: first, optimize its interface with environmental inputs [203,204] as well as its responding behavior. Second, optimize its internal organization so that the input can be mapped into the output as cheaply as possible. These optimizations entail minimizing metabolic costs, e.g., from heat dissipation or entropy production. Some of these costs depend on information theoretical input–output relationships, and they are detached from the material implementation of a circuit [35,36]. Ideally, we would write down all of the thermodynamic ingredients involved to calculate detailed balances of computations happening in candidate neural structures. This would reveal what wiring patterns are likely (i.e., more optimal in the eventual cost–benefit balance), given a fixed set of computations to be implemented (i.e., sought cognitive phenotype), energetic affordances and demands, and entropic losses.

Performing such comprehensive calculation is unrealistic, even for simple duplicated structures, on which this paper focuses. Our showcase of examples from real brains illustrates that some relevant aspects of the problem depend tightly on the specific behavior implemented by each circuit. Hence, actual thermodynamic potentials for this problem may rely largely on the circuit’s computational ends. But clever abstractions might reveal outstanding dimensions that are common to diverse phenotypes. Finding such coarse-grained dimensions would allow us to build effective cost–benefit balances and phase diagrams. The mathematics of arbitrary, emergent cost-benefit tradeoffs map back to the formalism of statistical physics [183,184,185,186,187]. Thus, neural structures might still be captured by such phase diagrams and “phase” transitions (akin to thermodynamic ones) might be uncovered.

We work out a simple case explicitly to study subcircuits within a neural structure before and after they cooperate towards a complex cognitive phenotype. Our cost–benefit calculation reckons: (i) the fitness benefit garnered by the implemented computations, (ii) expenses to coordinate redundant circuits, and (iii) other costs associated to normal (i.e., non-redundant) computing. Ultimately, all costs originate in thermodynamics (e.g., metabolism or material expenses). In the resulting diagrams:Two phases exist for uncooperative preconditions: one with a single structure and another with a duplicated structure. Large coordination costs (e.g., because the structures are far apart) result in a single structure. This supports that functions must lateralize due to enlarged brains [139].An additional third phase appears in cooperating preconditions. In it, the complex phenotype cannot emerge. Furthermore, the diagram is distorted, so each phase happens for different parameters than before.Superposing both diagrams shows evolutionary paths for the emerging phenotype, including:
₋A reliable path that results in the duplication of single neural structures. This has been proposed as a frequent mechanism for unfolding cognitive complexity [188].₋The absence of a direct route from duplicated to single structures. This suggests that the emergence of novel function cannot prompt lateralization (e.g., as in language) with the elements in our cost–benefit study alone.

In a second example, we speculate about how duplicated neural structures might further evolve once they are in place. This may depend on the specific phenotype that the structures implement and on how it expands the cognitive landscape. We propose three salient dimensions that might constrain evolutionary paths: (i) complexity of input signals ($K_{IN}$), (ii) complexity of sought output behavior ($K_{OUT}$), and (iii) complexity of the input-output mapping ($K_{MAP}$). Guided by them, we elaborate a tentative morphospace where we attempt to locate some examples reviewed in Section 2. We expect that reactive and predictive brains are dominated by high input and output complexity, respectively. We also note that structures with high, yet balanced $K_{IN}$ and $K_{OUT}$ (namely language and sensory-motor centers) have evolved separated regions specifically devoted to input processing and output generation. However, evolutionary evidence from early mammals also suggests that motor control was originally handled by somatosensory centers [138,139,140], potentially suggesting that the input–output segregation did not happen until motor control exceeded some complexity threshold (which might be captured by $K_{MAP}$ reaching a critical value in our morphospace). While these conjectures are speculative, the morphospace can advise numerical simulations to clarify these points.

Both examples presented have been inspired by concepts from Statistical Physics and Information Theory. We think that Statistical Physics (with its phase transitions and diagrams, criticality, susceptibilities to external parameters, etc.) is a very apt language to study the optimality and abundance of neural structures. Perhaps it is an unavoidable language. The task is big, but efforts are building up [5,35,36,37,39,48,49,56,60,61,62]. We hope to see important developments soon, hopefully along empirical records of neural circuitry falling within the resulting phase diagrams.

## Figures and Tables

**Figure 1 entropy-22-00928-f001:**
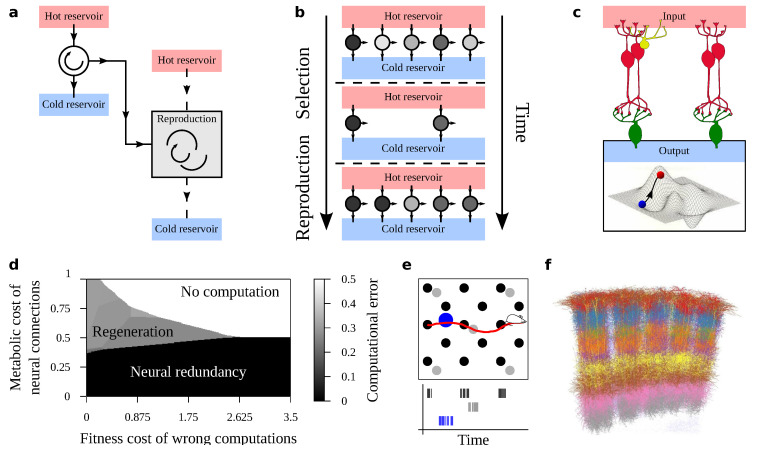
Towards a statistical physics of neural circuits. (**a**) Some thermodynamic engines can use free energy to shape matter as a replica of themselves. (**b**) Such self-replicating patterns can kick-start Darwinian evolution, deeply altering the optimality of different phases of matter. Efficient replicators are favored and their internal arrangement become subject of detailed optimization. (**c**) Neural circuits enable complex cognition in self-replicating patterns. Accordingly, they become subjected to evolutionary-thermodynamic pressures eventually measured by their performance in a computational landscape. Varying external parameters (e.g., computational task, energetic constraints …) renders phase diagrams for neural architectures. (**d**) Phase diagram derived from [37]. Neural circuits recover from injuries by using (i) costly redundant connections or (ii) paying a metabolic cost for regeneration. Three phases emerge: two where either one of the two strategies is preferred, and one where the computational phenotype never emerges. Transitions between phases constraint evolutionary paths. (**e**) Grid cells span the available room with shorter (black) or larger (gray) periods to create exhaustive spatial representations. Place cells (blue) encode specific locations. The spiking of each cell type is shown along the mouse’s trajectory. (**f**) Reconstruction of archetypal cortical columns by Oberlaender et al. reconstruction methods in [38].

**Figure 2 entropy-22-00928-f002:**
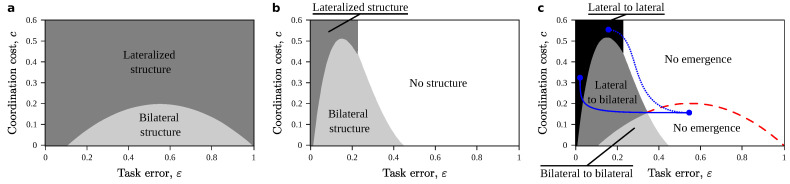
Phases of single and duplicated neural structures. (**a**) Phase diagram shows parameter combinations where single (darker) or duplicated (lighter) are preferred when subcircuits are not cooperating towards an emerging phenotype. (**b**) Phase diagram when subcircuits cooperate towards an emergent phenotype. A new phase exists, in which no circuit is implemented at all (white). (**c**) Superposing both phase diagrams reveals evolutionary paths for the emergence of the complex phenotype from pre-adaptations. Depending on metabolic costs and fitness contributions, its emergence might be blocked (white) or demand that a single circuit gets duplicated. Blue paths indicate necessary changes in external (metabolic and efficiency) conditions for an evolutionary path that takes us from a bilateral to a lateralized structure.

**Figure 3 entropy-22-00928-f003:**
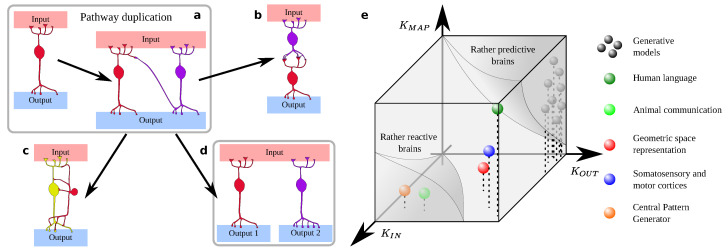
The garden of forking neural structures. ((**a**) Two neurons become duplicated: how do they evolve further? (**b**) Might one become specialized in input processing and another one in output control? (**c**) Might one take care of I/O interface and the other one become a controller that allows for subtler response? (**d**) Might they split ways, effectively dividing the original task, or perhaps exploring new functions? (**e**) The answers might depend on broad aspects of the task at hand, such as its input or output richness, or intrinsic mathematical complexity. In this speculative morphospace we attempt to locate some cases reviewed in Section 2. Separation of reactive vs predictive brains is based on their dependence on input and output richness, respectively. Predictive brains need to contain generative models (gray balls), circuits capable of generating context-dependent representations richer than the input that elicited them.

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
