# Peer review of "Fate of Duplicated Neural Structures"

_entropy, 2020, doi:10.3390/e22090928_

Round 1
Reviewer 1 Report
Review of "Fate of Duplicated Neural Structures"
Decision: Accept with Moderate Revisions
Overall, this is an excellent paper which combines literature review with computational modeling to produce an innovative manuscript. There are a few areas in which concepts and results are unclear, but otherwise the manuscript is easy to understand. This work also raises a number of interesting questions which may be lead to future research. But as a self-contained research unit, the manuscript is quite sound.
That being said, I have a number of comments that I would like to see address before acceptance:
General:
Why did you choose to use the word "fate" in the title? Is this analogous to the cell differentiation process? The rationale should be mentioned in the Introduction.
A few examples of informal sentences (ones that end with : should be made more formal).
Section 1:
The first part of the Introduction does not seem to fit well with the rest of the paper. Figure 1 looks more like a summary figure.
Good decision to limit the scope of the paper, but first few paragraphs seem to presage something big.
* last paragraph of Part 1: if duplications appear as a phase transition, and as a consequence of many distinct processes, is it suitable to characterize as a phase transition?
* how does this fit into the idea of phenotypic modules?
* is bilateral symmetry a phase transition? Is it the same (qualitatively) as a duplicated circuit? Approaching this as symmetry breaking deals with this issue some, but is all symmetry equal, particularly in terms of breaking.
Section 2:
This is a nice summary of duplication across a number of systems. Aside from being disparate structures, duplication is a common strategy of neural systems.
* like the discussion in 2.6, especially the distinction between reactive and predictive brains. The lead in to individual questions in this section could tie the concepts of duplication and symmetry-breaking together a bit more.
* The gedenken experiment in 3.2 could be a bit clearer in establishing why this is happening. What is the relevance -- while it is understandable after you read it, please make explicit at the beginning of this section.
Figure 3:
What are the generative models? Are they hypothetical functions of the brain? If this is inded correct, please clarify in the legend.
* a follow-up point from earlier in the paper: is duplication at different scales (neurons vs. circuits) equivalent? The morphospaces discussion answers this somewhat -- but this should be mentioned and made explicit in the Introduction.
* in the legend: reactive and predictive brains can be distinguished by an input self-similarity dimension. Are there other ways to more precisely determine scale between different types of architecture? I do not understand how you made the distinction between the two groups from your morphospace. Making this a bit more clear is crucial.
Section 4:
Second paragraph: "Deep down" can be changed to "Ultimately". The rest of the section is written quite informally.
- summarize what was found in a table, and summarize the results taken from the methods. It is not clear to me, although you did make those connections, they might need to be unpacked a bit.
--------------------------------------------------------------------------
Addition comments: (reply to author by email)
In the comment you highlighted, I was referring to bringing together the points raised in Section 2.6 and elsewhere into discussion of symmetry-breaking and duplication, preferably in the Introduction. Executed in the paper, this would ideally be a more upfront discussion of these topics in the Introduction, followed by reference to this in later sections. That way, the reader can have a frame of reference for these concepts as soon as possible in the paper. If the authors have a better idea on how to do this, I look forward to reading about it in the revision. Any other questions, let me know.
--------------------------------------------------------------------------
Author Response
Dear Referee:
Please, find my comments enclosed in the .pdf document attached.
Best regards:
The author

Reviewer 2 Report
This paper discusses the conditions under which duplications of neural circuits is beneficial, arguing for phase diagrams as functions of various control parameters.
This was a very pleasurable read, but it almost felt like a Perspectives piece that argued for a particular research program. I am glad to see that more people are arguing for research programs like this, especially since it pays homage to Horace Barlow's seminal efficient coding hypothesis from decades ago. However, I have to judge this article as a research report. As such, I'm concerned because I felt like the one numerical cost/benefit analysis was unclear and at odds with the usual assumptions in the field. If the authors can make their simple analysis clearer, I will happily recommend publication.
My issue with the simple example was largely that I could not understand it. I wasn't clear on if the K^0 tasks were all the same or all different, or some the same and some different. My best reading of this is that all the tasks were different and that some of the tasks remained undone after they had been divvied up, but then a later sentence on raising the benefit per subtask to a power of K^0 confused me. Aside from that:
-- It was very jarring to see the authors assume that two independent answers might interfere and provide a cost rather than a benefit. Decades of neuroscience research suggest that the brain knows how to error correct!
-- It might be nice to explain what b and c might physically be, e.g. is one of them related to energy costs of processing, which can be found in Atwell and Laughlin? Is benefit related to the energy obtained by doing the computation?
-- I think it's worth justifying why reliability epsilon is linearly related to cost C. I don't usually think reliability is so cheap, especially not in engineered devices.
-- If there's only one utility function, instead of a constrained optimization problem, then I'd recommend explaining why b and c have the right units to put the benefit and cost on equal footing.
-- Later, benefit per task is raised to a power to get the benefit total, and the explanation is that each subtask must be properly implemented for the organism to function. If benefit were actually code for "probability that subtask is completed" and all these probabilities were independent, then this benefit total would make some sense, but the benefit per task is certainly not a probability. I think it might make the entire section more readable if the authors compute probabilities of all tasks being completed or not and assign a large benefit to completion and a large cost to lack of completion.
I hope I've conveyed my confusion properly. I think more explanation of assumptions would allow the readers to better follow the math. I'm eager to see how the authors clarify their example.
The comments below are mild comments on various statements that deserve some care or typos:
-- Line 25: "of radically"
-- Line 33: "(Figure 1b" missing paren
-- Line 34: I would not say organisms often operate close to thermodynamic limits. Maybe given computation, they do, but you should definitely add in "given computation" to make this less of a lightning rod.
-- Line 54: RCs can be highly redundant depending on your reservoir recipe.
-- Line 63: "out our"
-- Line 91: people usually invoke plasticity to explain this. There were likely not duplicated structures pre lesion.
-- Line 108: "result detrimental"
-- Line 124: Could point out that there might be duplications within a single hemisphere's circuitry, in that the same neural network is duplicated.
-- Line 269: I'd be careful of this statement. Just by dint of being a physical system, protozoa are capable of quite a bit, in theory.
-- Line 326: extra paren
Author Response

(The authors gave the same response as above.)

Round 2
Reviewer 1 Report
This version of the manuscript looks like all of my comments have been incorporated and I am ready to approve an accept.
Reviewer 2 Report
The minimal model has now been adequately explained, and I recommend publication.
I have one more recommendation. I'm not completely sold on how the minimal model is set up. In particular, I'm not sure I would have set it up so that the cooperative benefits should only arise if all subtasks are correctly completed. One of the benefits (it seems) of having two different brain structures that can do duplicate computations is that success is an OR rather than an AND, so that the expected benefit should increase, but at the cost of expensive additional machinery to combine two different streams of information correctly. I think reasonable people can disagree on the ingredients in a minimal model, but as some readers might come in with similar intuitions to mine, it might be worth checking for the readers if the expected benefit of the cooperative scenario is sometimes less than the expected benefit of the independent scenario, and if it is, explaining more why you chose that. For example, if you had a particular model system in mind, maybe explaining how your minimal model assumptions align with your model system might help readers like me.